# Scalable Wettability Modification of Aluminum Surface through Single-Shot Nanosecond Laser Processing

**DOI:** 10.3390/nano13081392

**Published:** 2023-04-17

**Authors:** Chi-Vinh Ngo, Yu Liu, Wei Li, Jianjun Yang, Chunlei Guo

**Affiliations:** 1GPL Photonics Laboratory, State Key Laboratory of Luminescence and Applications, Changchun Institute of Optics, Fine Mechanics and Physics, Chinese Academy of Sciences, Changchun 130033, China; chivinh@ciomp.ac.cn (C.-V.N.); liuyu173@mails.ucas.ac.cn (Y.L.); weili1@ciomp.ac.cn (W.L.); 2University of Chinese Academy of Sciences, Beijing 100049, China; 3The Institute of Optics, University of Rochester, Rochester, NY 14627, USA

**Keywords:** single-shot nanosecond laser irradiation, wettability conversion, hydrophobicity, superhydrophobicity, short time fabrication

## Abstract

Conversion of a regular metal surface to a superhydrophobic one has great appeal because of the wide range of potential applications such as anti-fouling, anti-corrosion, and anti-icing. One promising technique is to modify surface wettability by laser processing to form nano-micro hierarchical structures with various patterns, such as pillars, grooves, and grids, followed by an aging process in the air or additional chemical processes. Surface processing is typically a lengthy process. Herein, we demonstrate a facile laser technique that converts the surface wettability of aluminum from inherently hydrophilic to hydrophobic and superhydrophobic with single-shot nanosecond laser irradiation. A single shot covers a fabrication area of approximately 19.6 mm^2^. The resultant hydrophobic and superhydrophobic effects persisted after six months. The effect of the incident laser energy on the surface wettability is studied, and the underlying mechanism of the wettability conversion through single-shot irradiation is suggested. The obtained surface shows a self-cleaning effect and the control of water adhesion. The single-shot nanosecond laser processing technique promises a fast and scalable method to produce laser-induced surface superhydrophobicity.

## 1. Introduction

Controlling the wettability of metal surfaces to hydrophobic or superhydrophobic has attracted a lot of attention because the instinct hydrophilic property of metal surfaces causes critical problems in practical applications. For example, corrosion is an important issue for metals for industrial productivity and the natural environment [1,2,3] because metal surfaces may come into contact with water or work in a humid environment for a long period of time. In order to prevent a metal surface from corrosion, a protective barrier of paint is commonly used to shield the metal surface from the corroding agents [4,5]. However, paints and coatings usually do not last long, and the peel-off will re-expose the metal surface to the environment. A superhydrophobic metal with its unique self-cleaning effect can be a solution.

Recently, lasers have been used to modify and probe the wettability and related properties of metal surfaces [6,7,8,9,10,11,12,13,14,15]. These techniques are simple, precise, and more sustainable than paints or coatings. First, a focused pulsed laser is used to produce nano-micro hierarchical structures on the metal surface with various patterns such as pillars, grooves, and grids. To create these structures, a high repetition rate laser is often used [16,17,18,19,20]. Quite often, the conversion of a metal surface to hydrophobic or superhydrophobic also requires an aging process or chemical coating [21,22,23,24,25,26,27]. For example, Jagdheesh et al. used a nanosecond laser with a repetition rate of 100 kHz to create a micro-hole array on aluminum surfaces and put the samples in the ambient air for an aging process [21]. The samples converted to hydrophobic after 24 h and to superhydrophobic after a few weeks. This phenomenon happened in the same manner as that in our publication on femtosecond-treated platinum [10]. To shorten the wettability transition time, we recently employed an additional heat treatment process after the femtosecond laser ablation on Al and various metals [28]. The wettability transition time was shortened from a couple of weeks to within 30 min. The mechanism for this rapid wettability transition was also proposed. The main application for this research was for anti-corrosion use. However, our previous studies and the current fabrication approaches have used a high repetition rate (the overlapping of many single shot pulse lasers) to produce nano-micro hierarchical structures, which might be a slow process for scalability.

In this research, a facile technique using single-shot nanosecond laser irradiation was introduced, with a fabrication area of approximately 19.6 mm^2^ for a single shot, which is larger than other studies’ fabrication areas with a single shot by 27 to 110,000 times [29,30,31,32,33,34,35]. The nanosecond laser was employed because it is compact, robust, cost-effective, and suitable for real industrial applications. The wettability of aluminum was converted from hydrophilic to superhydrophobic in a short time. We showed that this superhydrophobic effect persisted over six months. The effect of laser energy on surface wettability was studied to find the optimum laser energy which can provide good superhydrophobicity. The mechanism for wettability conversion from inherently hydrophilic to hydrophobic or superhydrophobic using only one single-shot irradiation was explained by surface morphology and surface chemistry. Moreover, the superhydrophobic aluminum proved its self-cleaning effect and control of water adhesion. The obtained laser technique, which showed wettability conversion on aluminum as presented in this research, and on stainless steel as partly presented in the Appendix A, could open a simple and scalable approach to produce hydrophobic or superhydrophobic metal.

## 2. Materials and Methods

### 2.1. Fabrication Process

A Nd:YAG nanosecond pulsed laser system (Q-smart 850, Quantel, Newbury, UK) was used for irradiation on the metal surface. The laser system is shown in Figure 1. The nanosecond laser head is controlled by a power supply and a Q-touch control panel. In the experiment, the single shot mode (only shooting one pulse for one time) was adopted. The laser shot beam was guided by the reflection mirrors to a sample of commercial aluminum sheets (Al 115905 99.999%), which was put on a stage. Before the laser processing, aluminum sheets were wet-polished with sandpapers of 400, 800, 1200, 1500, 3000, 5000, and 7000 grits by a precision lapping/polishing machine (UNIPOL-802, MTI Corporation, Richmond, CA, USA). During the experiment, the influence of the laser energy (9, 18, 38, 73, 135, 192, 254, and 320 mJ) on the surface wettability of aluminum material was investigated. Moreover, this technique was also tried on stainless steel with different laser energies, and the data of wettability change on this material are shown in the Appendix A.

### 2.2. Surface Characterization

The surface morphology of the sample after the single nanosecond pulse laser processing was investigated by scanning electron microscopy (SEM, Phenom ProX, Phenon World, Eindhoven, The Netherlands). X-ray diffraction (XRD, D8 FOCUS, Bruker, Billerica, MA, USA), point energy-dispersive X-ray spectroscopy (EDS, Phenom ProX, Phenon World, Netherlands), and Fourier-transform infrared (FTIR, Cary 630 FTIR, Agilent Technologies, Santa Clara, CA, USA) were used to evaluate chemical compositions on the surface. The wettability on the laser-treated surface was measured by an optical contact angle and interface tension meter (SL200KB, Kino, Kailua Kona, HI, USA) with a 5-μL volume of deionized (DI) water droplets.

## 3. Results

### 3.1. Change of Surface Morphology

Figure 2 shows the morphology of the aluminum surface after the irradiation of only one nanosecond laser pulse. It is clear that after the laser processing, some white particles appeared on the aluminum surface in a random manner. The surface had an average dimension of 555 nm within the range from 86 nm to 1240 nm. The distribution density of these nanoparticles was approximately 89 particles/100 μm^2^. Moreover, after the laser-material interaction, the initial grey color of the surface became brighter when increasing the laser energy, as shown in the top-left small-captured image in Figure 2. The formation of the white nanoparticles was also found when irradiating on the surface of stainless steel, as shown in Appendix A. However, the average dimension of the white nanoparticles on stainless steel is smaller than that on aluminum, which was approximately 289 nm within the range of 62 nm to 712 nm. In this case, the nanoparticles were randomly distributed on the surface, with a density of 86 particles/100 μm^2^. In general, after the irradiation of one nanosecond laser pulse onto the targets, the nanostructures and the sub-microstructures were formed on the surface. Of course, the size of the surface structures depends on the material properties.

### 3.2. Change of Surface Compositions

After the wet polishing of the pure aluminum surface, the oxygen content was increased. This comes from the formation of an oxide layer on the aluminum during the polishing process, which is in agreement with another study [36]. Chemical compositions were investigated on the white nanoparticles or sub-microparticles of the laser-treated aluminum surface, and on the wet-polished aluminum surface before the laser processing using an EDS measurement, as shown in Table 1. With respect to the wet-polished aluminum surface before the laser processing, the ratio between oxygen and aluminum content on the white nanoparticles or sub-microparticles exhibited a higher value. Significantly, the amount of carbon content on the sample surface increased more than two times immediately after the laser processing. It appeared to continue to rise over the following six months.

As shown in Figure 3a, the XRD measurements showed that the crystalline structure of the sample had no change after the laser irradiation. However, there was a change in chemical compositions from the EDS measurement. Therefore, the increment in the carbon content may come from another factor. Additional FTIR measurements on the sample surface before and after the laser processing were carried out, as shown in Figure 3b. It was determined that after the laser processing, some new peaks appeared, which were characterized by the chemical bonding of the hydrophobic groups (–CH_3_ and –CH_2_–). Because such medium-strong hydrophobic groups have an inherently low surface energy, the wettability of the aluminum surface is improved. The increment of carbon content as well as the appearance of new organic groups can demonstrate that the new hydrophobic bonding or the new hydrophobic layer was formed after the laser processing.

### 3.3. Surface Wettability Conversion

Figure 4a shows the measured contact angles of the sample surface before and after irradiating one single nanosecond laser pulse with an energy of 192 mJ. The contact angle of the sample after the single-shot nanosecond laser irradiation at the pulse energy of 192 mJ changed from 80° to 160°, which indicates the wettability transition from a hydrophilic to a superhydrophobic property. In addition, the sample showed a sliding angle of 9°, as shown in Appendix A. The surface treatment at a pulse energy of 192 mJ was the optimal result in the study of the laser energy’s effect on the surface wettability, as shown in Figure 4b. When the laser energy increased, the contact angle also increased from inherently hydrophilic (80°) to hydrophobic (greater than 90°), and finally to superhydrophobic (greater than 150°). The superhydrophobic state of the treated sample reached the highest contact angle at a pulse energy of 192 mJ. However, when the laser pulse energy increased above 192 mJ, the contact angle slightly decreased, but still maintained the superhydrophobic state (i.e., the contact angle was still greater than 150°). After one month had passed after the laser processing, the contact angles of all the treated samples were re-measured. The contact angles did not change significantly when the laser irradiation energy was smaller than 192 mJ. At pulse energies greater than 192 mJ, the contact angle increased significantly, and a 5-μL of DI water droplet could not be placed on the laser-exposed areas. Therefore, the enhancement of superhydrophobicity happened continuously after laser processing, especially for the samples treated at energies greater than 192 mJ. This statement was also demonstrated more clearly when measuring the contact angles of all treated samples again after six months after the laser processing. The contact angle on the laser-treated area with 135 mJ increased significantly, and a 5-μL of DI water droplet could not be put on the laser-exposed area of the sample. The reason is that the surface energy of the laser-exposed area became lower than before, which made the water droplet more challenging to separate from the needle to the surface. In addition, six months after the laser processing, other samples with laser energies less than 135 mJ also increased their contact angles at the laser-treated areas, which indicates a reduction in the surface energy over time after the laser processing.

Interestingly, the laser-treated area with the laser energy of 9 mJ converted from hydrophilicity to hydrophobicity six months after the laser beam interaction. Therefore, after the interaction between the single laser pulse and the aluminum target, the material surface became hydrophobic or superhydrophobic, depending on the treated energy. This hydrophobic or superhydrophobic property has continuously enhanced itself over time since the laser irradiation. Appendix A illustrated the wettability conversion before and after the laser processing, as well as the enhancement of superhydrophobic properties after one month and after six months since the laser irradiation. An impact test with water droplets was performed on the sample just after laser ablation, as shown in Appendix A. After dropping 18 droplets, a small amount of water started to adhere to the surface. Although the surface became superhydrophobic just after laser ablation, the superhydrophobic performance was not high, which was demonstrated through the contact of the droplet on the surface in Appendix A and the droplet impact test in Appendix A. In addition, the wettability conversion on Al, the surface wettability conversion from hydrophilicity to hydrophobicity after irradiating with a single-shot laser beam, was also found on stainless steel, as shown in Appendix A. However, its hydrophobic property was not stable, but the contact angles persisted at certain values higher than the initial value of the wet-polished surface before the laser irradiation. The reason for this instability as well as the mechanism for wettability conversion on stainless steel might be investigated in further research.

### 3.4. Mechanism of Wettability Conversion with Single-Shot Nanosecond Laser Irradiation

After the laser processing, both the morphology and the chemical compositions of the aluminum surfaces changed, which could provide the mechanism for the wettability conversion from hydrophilicity to superhydrophobicity. When a single laser pulse interacted with the aluminum surface, laser energy was absorbed on the surface. At that time, thermal effects on the laser-treated area occurred [37,38,39]. The estimated maximum temperature was about 546 °C, as calculated by thermal models [38,40]. The existing oxide layer on the aluminum surface, which was formed by the wet-polishing process, might be modified and accumulated as nanoparticles and sub-microparticles. At the same time, the pure aluminum layer underneath the existing oxide layer was also oxidized, and formed nanoparticles and sub-microparticles due to the nanochemical effects of the pulsed laser interaction with metals [41]. All of the formed particles could act as nanostructures and sub-microstructures on the aluminum surface. These structures included a higher oxygen content compared to the initial oxygen content in the aluminum surface before the laser processing. The oxygen content might indicate the appearance of aluminum oxide as well as the appearance of a hydroxyl group (–OH) on the aluminum oxide. As an effective adsorptive site, the hydroxyl group could adsorb the organic matters from air moisture [28,42,43]. The organic matter in the air could be acetic acid or formic and polymeric hydrocarbons [44,45,46], which can include short nonpolar or hydrophobic molecules, and thus decrease the surface energy. Once the organic matter reacted with hydroxyls, their nonpolar functional groups could be chemisorbed onto the formed nanostructures and sub-microstructures, leading to the reduction of surface polarity or surface energy. The corresponding chemical reaction can be described in Equation (1) [47]. While the sign ≡ is a triple bond, R represents the organic matter in the air. The proposed chemisorption mechanism could be examined by the obtained results in the EDS measurement with the increment of carbon content and oxygen content as well as by the obtained results in the FTIR measurement with the appearance of the hydrophobic bonding (–CH_3_ and –CH_2_–). The organic adsorption phenomenon would continue, and over time, the hydrophobicity and the superhydrophobicity of the aluminum surface would occur, as demonstrated in the results of the measured contact angles after one month and after six months since the laser irradiation.
(1)$$\equiv AlOH+RCOOH⇆\;\equiv AlOOCR+H_{2}O$$

In summary, as shown in Figure 5, after the laser processing on the aluminum surface, many unsaturated metal and oxygen atoms were created as a form of nanostructure and sub-microstructures with the aid of thermal effects. At the same time, chemisorption also occurred. Compared to a femtosecond laser, a nanosecond laser has a longer laser action time on the surface. Therefore, the formation of the nano/sub-microstructures and the chemical reactions could happen in a single shot because the thermal effects are prolonged. When the laser energy increases to a critical point, the thermal energy also rises to a critical point [38,48]. The increment of thermal energy can accelerate the structure formation and the hydroxylation reaction. Depending on the laser energy, the adsorbed amount of hydrophobic bonding (–CH_3_ and –CH_2_–) from the air might be different, which can be represented by the results of contact angle measurement with the change of laser energy. The combination of nanostructures and sub-microstructures and their low surface energy from the adsorbed hydrophobic bonding can make the aluminum surface hydrophobic or superhydrophobic. Moreover, the chemisorption still occurred after the surfaces became hydrophobic or superhydrophobic in order to enhance their hydrophobicity.

### 3.5. Potential Applications

Hydrophobic metal, especially superhydrophobic metal, can have many potential applications, such as self-cleaning, anti-corrosion, anti-icing, water transportation, etc. because water cannot touch the metal surface, or it can easily be rolled off the metal surface with a small tilting angle. For example, sliding water droplets were able to clean a 30°-tilted superhydrophobic Al surface covered in cellulose powder, while they could not clean those on a 30°-tilted non-treatment Al surface (Figure 6a and Appendix A). In addition, when applying different laser-exposed energies on Al surfaces, their hydrophobicity is different due to different surface energies, which can suggest the control of water adhesion for water transportation like a “droplet tweezer” and liquid mixing (Figure 6b,c, and Appendix A). A concept of transporting a micro-droplet array called probe droplets to form the mixtures with target droplets can be suggested for potential bio-applications, as shown in Figure 6d.

From the technological transfer point of view, when coordinating a single pulse laser with the help of the translation system for practical scalable wettability modifications at millimeter- to meter-levels, some areas may not be treated by the laser beam because the trace of the laser pulse is circular. On the other hand, overlapping laser pulses may create non-uniformity in superhydrophobic performance between the overlapped areas and the non-overlapped areas due to different treated laser energies. To ensure that only the desired areas become superhydrophobic without the laser-pulse overlapping, a masking approach can be used to combine with the single-shot nanosecond laser processing and translation systems, enabling a high potential application of our proposed technology.

## 4. Conclusions

When irradiating only a single nanosecond laser pulse with sufficiently defined energy, the aluminum surface could convert from inherently hydrophilic to superhydrophobic, with an irradiated area of approximately 19.6 mm^2^, which is a much larger exposure area compared to those of other conventional focused laser beam processes. The effect of the laser energy on the surface wettability was investigated to find the optimal laser energy at 192 mJ in this research. Moreover, the mechanism for the wettability conversion using one single-shot nanosecond laser was proposed by a combination of the nanostructures and sub-microstructures, and the low surface energy of the hydrophobic adsorbed bond (–CH3 and –CH2–) with the aid of thermal effects on the surface. The resultant hydrophobic or superhydrophobic properties can persist beyond six months. Additionally, the laser-treated surfaces demonstrated their feasibility in self-cleaning and controlling water adhesion applications. The presented nanosecond laser technique is simple, cost-effective, and available for scalable wettability modification, which can be useful in various industrial applications.

## Figures and Tables

**Figure 1 nanomaterials-13-01392-f001:**
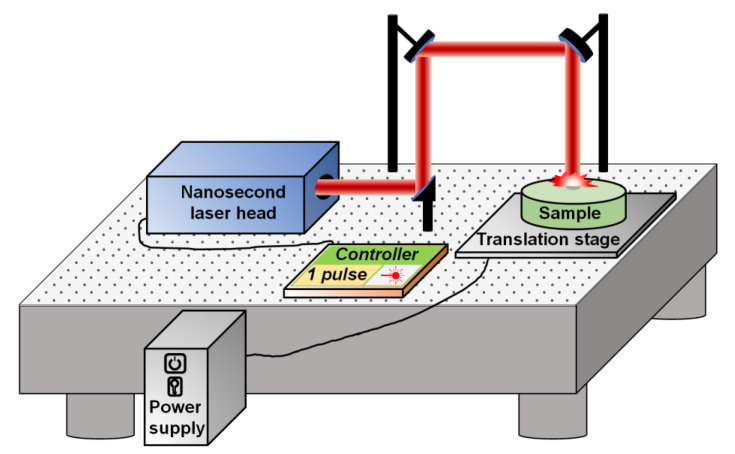
Experimental setup of single-shot nanosecond laser processing on metal surfaces.

**Figure 2 nanomaterials-13-01392-f002:**
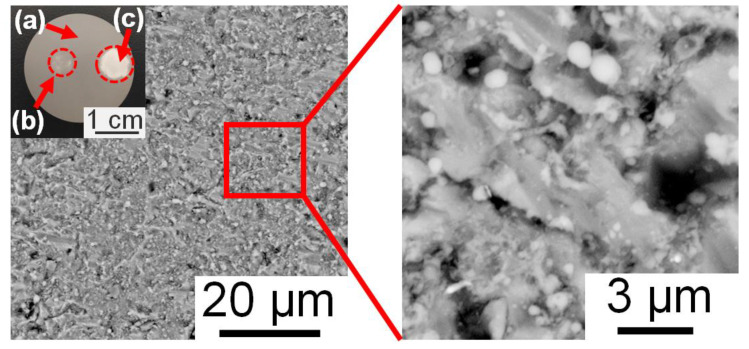
SEM images of the aluminum surface after the single-shot irradiation with the energy of 192 mJ. Note that (**a**) is the area before laser processing, (**b**) is the low-laser-energy treated area, and (**c**) is the high-laser-energy treated area.

**Figure 3 nanomaterials-13-01392-f003:**
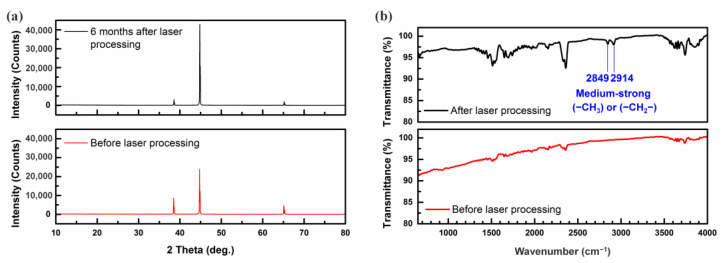
(**a**) The XRD result, and (**b**) the FTIR result of the aluminum surface before and after the single-shot irradiation at 192 mJ.

**Figure 4 nanomaterials-13-01392-f004:**
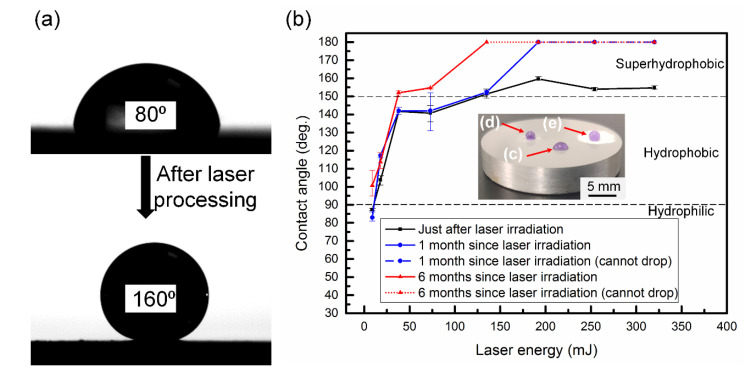
(**a**) Measured surface wettability of the aluminum target after the single-shot nanosecond laser irradiation at the pulse energy of 192 mJ; (**b**) Measured surface wettability of the aluminum target after the laser processing, after 1 month and 6 months since the laser processing with different laser energies; (**c**) Water droplet on the area before the laser processing; (**d**) Water droplet on the low-laser-energy treated area; and (**e**) Water droplet on the high-laser-energy treated area.

**Figure 5 nanomaterials-13-01392-f005:**
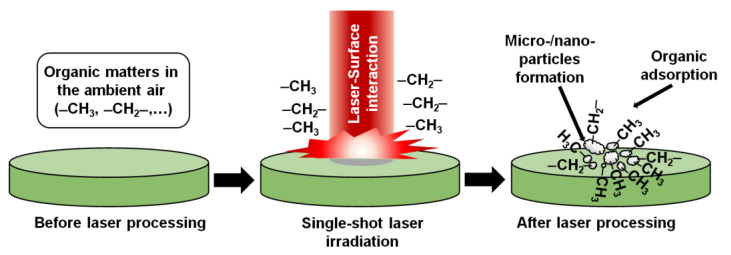
Mechanisms of the wettability conversion on the single-shot laser-treated aluminum surface.

**Figure 6 nanomaterials-13-01392-f006:**
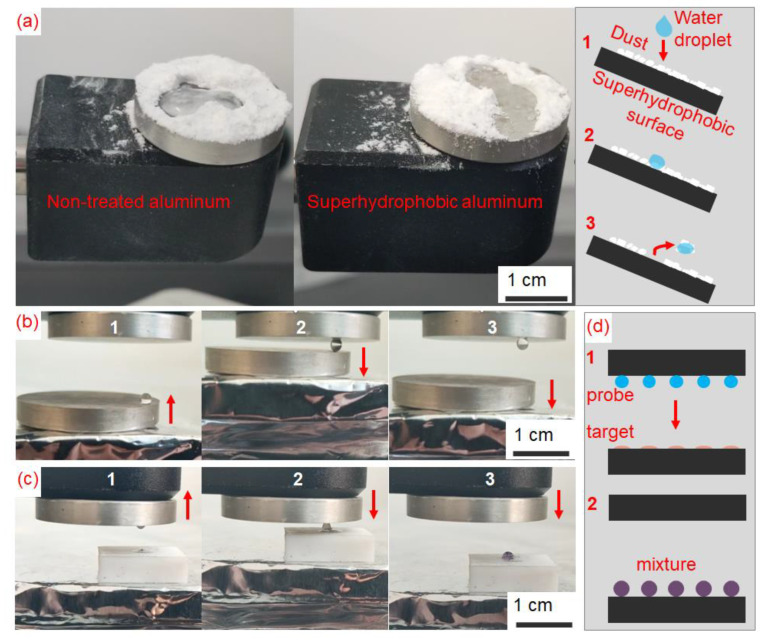
(**a**) Self-cleaning effect on non-treated and superhydrophobic aluminum surfaces with a describing schematic on the right side at three different time points, labeled as 1, 2, and 3 in sequential order (the red arrows showed the movement directions of the water droplet), (**b**) Droplet capture as a “droplet tweezer” on superhydrophobic aluminum with the different controlling level of water adhesion shown in three images taken from Appendix A at three different time points, labeled as 1, 2, and 3 in sequential order (the red arrows showed the movement directions of the lower linear translation stage), (**c**) Droplet mixing between a water droplet and an oil droplet shown in three images taken from Appendix A at three different time points, labeled as 1, 2, and 3 in sequential order (the red arrows showed the movement directions of the lower linear translation stage), and (**d**) a conceptual depiction of droplet array manipulation for droplet mixing applications with the initial state labeled as 1 and the final state labeled as 2 (the red arrows showed the movement directions of the upper linear translation stage).

**Table 1 nanomaterials-13-01392-t001:** EDS-measured chemical composition of the aluminum surface before and after the single-shot laser processing with the pulse energy of 192 mJ.

Element	Before the Laser Irradiation	Just after the Laser Irradiation	1 Month after the Laser Irradiation	6 Months after the Laser Irradiation
Al	63.80	55.60	52.73	51.00
O	32.40	36.19	34.15	32.80
C	3.80	8.21	13.12	16.20
O/Al	0.51	0.65	0.65	0.64
C/Al	0.06	0.15	0.25	0.32

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
