# Peer review of "Scalable Wettability Modification of Aluminum Surface through Single-Shot Nanosecond Laser Processing"

_nanomaterials, 2023, doi:10.3390/nano13081392_

Round 1

Reviewer 1 Report

The manuscript ID: nanomaterials-2321416 entitled: „Scalable wettability modification of aluminum surface through single-shot nanosecond laser processing” presents a technique to modify surface wettability of aluminium by laser processing.

The authors demonstrate that by using a facile laser technique converts the surface wettability of aluminum from inherently hydrophilic to hydrophobic and superhydrophobic with single-shot nanosecond laser irradiation. This effect is keeped for a long period of time (six months).

The authors should to take into consideration the following aspects:

1.      The same authors published a similar paper regarding the modify surface wettability of aluminium by using femtosecond laser processing and I consider that it is necessary to compare the obtained results by these two methods.

The contact angle of the flat aluminum surface was 49°,i.e. hydrophilicity. After femtosecond laser ablation,the contact angle became 0°,i.e. superhydrophilicity. The water droplet diffused suddenly on the just-after-laser-ablated surface to exhibit the contact angle of 0°[Opt express 2020, reference 41].

2.      After femtosecond laser ablation the aluminium surface become hydrophilic (0° contact angle) and by using single-shot nanosecond laser irradiation the aluminium surface become hydrophobic (150° contact angle). Something is not clear. Please explain.

Lines 208-210

As an effective adsorptive site, the hydroxyl group could adsorb the organic matter from air moisture [39-41], such as acetic acids, formic and polymeric hydrocarbons, which can include short nonpolar or hydrophobic molecules, and thus decrease the surface energy (this manuscript).

3.      Which is the evidence of the presence of acetic acids, formic and polymeric hydrocarbons in the air moisture?  

The mentioned references [39 and 40] do not show the presence of acetic acids, formic and polymeric hydrocarbons in the air moisture, only the reference [41] which is the reference of the same authors.

The corresponding chemical reaction can be described in Eq. 1.

≡ AlOH + RCOOH ⇆ ≡ AlOOCR + H2O

4.      In the equation 1 must to explain what is this sign ≡  and what can be R. In this form is incomplete the equation 1.

When a single laser pulse interacted with the aluminum surface, laser energy was absorbed on the surface. At that time, thermal effects on the laser-treated area happened, and consequently, melting, evaporation, or sublimation of the aluminum surface might occur.

5.      It is very difficult to suppose that after laser-treated aluminium surface appear melting, evaporation or sublimation, because the melting point of aluminium is 660°C, the melting point of Al2O3 is over 2000°C and only the melting point of aluminium hydroxyde is around 300°C.

6.      Which is the maximum temperature evolved by using of the single-shot nanosecond laser?

Author Response

Dear Reviewer 1,

Please kindly see the attachment for our responses.

Best Regards,

Reviewer 2 Report

The work is well organized, the subject is of interest, with an impact on possible industrial applications.

However, I have identified some points that have not been discussed and which are important for the technological transfer of the idea.

As can be seen from the microscopy images, the trace of the laser pulse is circular. With the help of the translation system, the coordinates of the laser irradiations can be controlled. However, considering that the traces of two consecutive pulses are tangent, what is the solution to modify the areas not affected by the laser, so that the superhydrophobicity is active at each point. This is very important from application point of view.

If the traces of the pulses are secant, so that the entire surface of interest is irradiated with the laser, how is the superhydrophobicity influenced in the areas where the laser pulses are overlapped.

I recommend that this manuscript to be published after minor revision.

Author Response

Dear Reviewer 2,

Please kindly see the attachment for our responses.

Best Regards,

Reviewer 3 Report

The authors conducted a study to transform the surface of aluminum from hydrophilic to superhydrophobic using only a single nanosecond laser pulse. In general, a larger area (19.6 mm^2) than other laser ablation studies was used in this study, and the wettability of the surface was analyzed according to the laser energy. As a result of this study, it was shown that the aluminum surface remains hydrophobic and superhydrophobic for more than 6 months.

The authors irradiated the sample with several intensities of laser energy, but data on morphology are lacking. In the results section, the authors only describe the analysis for a laser energy of 192 mJ. What is the fundamental reason for using 192 mJ of laser energy for analysis?

Also, the research background is lacking. Briefly introduce the methods and results of similar studies in the introduction section.

Author Response

Dear Reviewer 3,

Please kindly see the attachment for our responses.

Best Regards,

Round 2

Reviewer 1 Report

The authors acted on all comments of the reviewer intensively and revised their manuscript substantially, by adding new detailed discussion. With this the manuscript significantly improved in quality and can now be recommended for publication.